# 🤔 PALBERT: Teaching ALBERT to Ponder

**Nikita Balagansky, Daniil Gavrilov**
Tinkoff
n.n.balaganskiy@tinkoff.ai, d.gavrilov@tinkoff.ai

## Abstract

Currently, pre-trained models can be considered the default choice for a wide range of NLP tasks. Despite their SoTA results, there is practical evidence that these models may require a different number of computing layers for different input sequences, since evaluating all layers leads to overconfidence in wrong predictions (namely overthinking). This problem can potentially be solved by implementing adaptive computation time approaches, which were first designed to improve inference speed. Recently proposed PonderNet may be a promising solution for performing an early exit by treating the exit layer's index as a latent variable. However, the originally proposed exit criterion, relying on sampling from trained posterior distribution on the probability of exiting from the $i$-th layer, introduces major variance in exit layer indices, significantly reducing the resulting model's performance. In this paper, we propose improving PonderNet with a novel deterministic Q-exit criterion and a revisited model architecture. We adapted the proposed mechanism to ALBERT and RoBERTa and compared it with recent methods for performing an early exit. We observed that the proposed changes can be considered significant improvements on the original PonderNet architecture and outperform PABEE on a wide range of GLUE tasks. In addition, we also performed an in-depth ablation study of the proposed architecture to further understand Lambda layers and their performance.

## 1   Introduction

These days, fine-tuning pre-trained models on downstream tasks became a de facto standard technique for training NLP models (Devlin et al., 2019; Liu et al., 2019; Lan et al., 2020; Radford et al., 2019). Although, as model sizes increase, it becomes harder to use them for real world applications due to computational footprint. Because of this fact, practitioners may desire to perform an early model exit (i.e., with adaptive computation time (Graves, 2016)) to reduce the computation needed for a specific model input.

One model that is widely used in real-world applications is ALBERT (Lan et al., 2020), which is based on the Transformer architecture (Vaswani et al., 2017) with shared layers (i.e., the same layer is evaluated several times to provide an output). Sharing layer weights allows us to think of ALBERT as of recurrent model that simplify development of early exit methods. Most of them (Teerapittayanon et al., 2016; Liu et al., 2020; Xin et al., 2020) rely on performing an early exit based on entropy of predictions during the model evaluation. Although, Zhou et al. (2020) showed that running the ALBERT-Base block for a fixed number of times (10) could increase the accuracy of the fine-tuned model on specific tasks (e.g., MRPC) despite the entropy of predictions monotonously decreasing for all 12 layers. A recent PABEE solution (Zhou et al., 2020) was designed to overcome this issue by performing an early exit based on the consensus between different classifier heads from different layers. The model stops evaluating when several classifiers in a row produce the same result.

36th Conference on Neural Information Processing Systems (NeurIPS 2022).

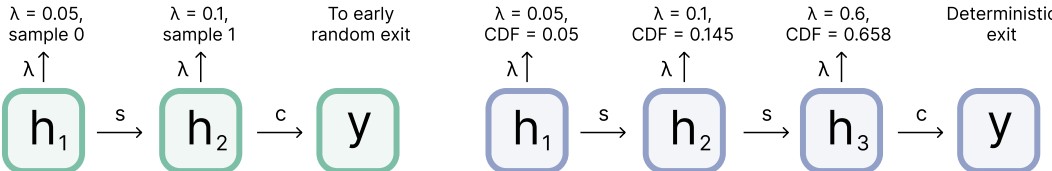

Figure 1: A comparison of the original sampling exit criterion of PonderNet (on the left) and the proposed Q-exit criterion (on the right). PonderNet performs sampling from the Bernoulli distribution obtained from the Lambda layer at each step, possibly exiting a model too early or too late. For Q-exit, we evaluate the Cumulative distribution function (CDF) of the probability of exiting at layer $i$. Once CDF becomes greater than the threshold value ($0.5$ in this example), we perform an early exit. With such a deterministic criterion, we can perform an early exit from a model more robustly without introducing variance in the exit layer's index during inference.

A possible orthogonal solution for this task is PonderNet (Banino et al., 2021) – a variational approach that treats the exit layer's index as a latent variable. By maximizing the lower bound of the likelihood of the training data, PonderNet trains a model which can predict whether it is necessary to exit from a specific layer during evaluation. However, Banino et al. (2021) proposed to sample from the trained posterior distribution of exiting from each layer during inference, which leads to major variance in exit layer indices.

This paper proposes improving PonderNet adapted for ALBERT (Lan et al., 2020) and RoBERTa (Liu et al., 2019) fine-tuning (PALBERT and PRoBERTa). Instead of performing an early exit by sampling from the trained posterior distribution during evaluation, we used a novel zero-variance exit criterion, namely **Q-exit**, which evaluates the CDF of the exit layer's probability distribution and performs a deterministic early exit. We also revisited the architectural choices of Lambda layers used to predict the probability of exiting from the current layer to make them aware of dynamics in hidden states across previous layers and the number of currently running layers.

We experimented with PALBERT and PRoBERTa on the GLUE Benchmark datasets (Wang et al., 2018). The ablation study showed that PALBERT produced significantly better results than the original PonderNet architecture adapted for ALBERT fine-tuning. Furthermore, the proposed methods outperformed PABEE and are comparable to plain ALBERT fine-tuning, while also exceeding it in speeds. We also analyzed the trained model and provided insights on further improvement of the variational approach for early exiting[1].

## 2   Related Work

Most of the approaches used to perform an early exit from a model are based on the probability distribution of predictions: BranchyNet (Teerapittayanon et al., 2016), FastBERT (Liu et al., 2020), DeeBERT (Xin et al., 2020), which can be seen as an entropy criterion. Xin et al. (2021) proposed BERxiT with LTE layer used to estimate the correctness of predictions during the inference based on the hidden state from the specific layer.

However, there is strong practical evidence that classification models' overthinking causes a reduction in predictions' entropy, making these methods difficult to use (Zhou et al., 2020). Furthermore, it is unclear how to adapt entropy methods for regression tasks (Zhou et al., 2020). Zhou et al. (2020) proposed PABEE – a method to perform an early exit based on several classifiers from the different levels of a model. Once several classifiers in a row (the number of these classifiers is determined by the patience hyperparameter $t$) produce the same result, we can perform an early exit. Several recent works also utilized the idea of a consensus-based exiting strategy (Zhu, 2021; Liao et al., 2021). Although, most of them mainly augment the basic consensus scheme with orthogonal heuristics. Because of this, we did not include them in our experiments and only used PABEE as a consensus-based method.

An alternative way to perform an early exit is the Ponder architecture (Banino et al., 2021), which uses auxiliary Lambda layers to predict whether a model should exit from a specific layer during

---

[1]Source code: https://github.com/tinkoff-ai/palbert

runtime. Inputs to Lambda layers used in PonderNet are hidden states from the current layer of a model. PonderNet can be seen as a model with the latent variable that corresponds to the exit layer index, which is trained by maximizing the lower bound of the marginalized likelihood of the data.

During inference, Banino et al. (2021) proposed to sample from the trained posterior distribution of exit layer probabilities. However, this exit criterion can lead to uncertainty in exit layers for the same input (see Section 4.5. Even if the Lambda layer produced a probability equal to $0.1$ of exiting from the first layer, we could still exit a model too early in one of ten cases, even though the probability was small. We also hypothesize that predicting exiting from a layer based entirely on a single hidden state could be sub-optimal since performing early exit could also depend on the dynamics in hidden states across layers (i.e., the Lambda layer should know how hidden states change during the evaluation).

## 3 Improving PonderNet

In this section, we describe the adaption of PonderNet to the ALBERT model (PALBERT). Although, it could be easily extended to other pre-trained models such as RoBERTa (which we also used in our experiments).

The usual ALBERT evaluation can be defined as a computation of $n$ hidden states $h_i = S(h_{i-1})$ from the input embeddings $h_0$ of an input sequence $x$, where $i \in [1; n]$. Once $h_n$ is obtained, it is passed to a classifier block $C(h_n)$ to get the parameters of an output distribution $p(y|x)$. A common way to fine-tune this architecture on downstream tasks is to initialize the embeddings and the $S$ layer by using ALBERT (pre-trained on Masked Language Modelling) while initializing $C$ randomly and then optimizing all parameters by maximizing the likelihood of the training data.

While plain ALBERT performs a fixed number of computational steps, it is possible to perform an arbitrary number of evaluations of the layer $S$. Banino et al. (2021) proposed to extend each Transformer layer with a so-called Lambda layer. More precisely, for each layer $i$, after $S$ outputs a new $h_i$, it is then passed to the classifier and Lambda layers to get parameters $C(h_i)$ of output distributions $p(y|x, i)$ and the probability of exiting from the $i$-th layer $\lambda_i = \Lambda(h_i)$, which induces a generalized geometric distribution on probability of exiting from layer $i$ equal to $p(i|x) = \lambda_i \prod_{j=1}^{i-1}(1 - \lambda_j)$.

Then, having the probability distribution from each layer $p(y|x, i)$, the parameters of the model are optimized to maximize

$$L(x, y) = \mathbb{E}_{i \sim p(i|x)}\Big[ \log\big(p(y|x, i)\big)\Big] - \beta KL\Big(p(\cdot|x)||p(\cdot|\lambda)\Big) \leq \log\big(p(y|x)\big) \tag{1}$$

Here, $p(\cdot|\lambda)$ is a prior geometric distribution of exiting from each layer, parametrized by the hyperparameter $\lambda$, and $\mathbb{E}_{i \sim p(i|x)}\Big[ \log\big(p(y|x, i)\big)\Big]$ is evaluated analytically by averaging likelihoods from different layers with posterior exit probabilities. If we treat the exit layer index as a latent variable, then optimizing $L$ from the Equation 1 could be seen as maximizing the lower bound of marginalized log likelihood $\log\big(p(y|x)\big)$ (Kingma and Welling, 2014).

Note that following Banino et al. (2021) the prior probability of exiting from the last layer $n$ is normalized as $p(n|\lambda) = 1 - \sum_{i=1}^{n-1} p(i|\lambda)$ in order to make $p(i|\lambda)$ sum into 1 with a finite number of steps, while $p(i|x)$ it truncated by the lowest layer index $j$, s.t. $\sum_{i=1}^{j} p(i|x) \geq 1 - 0.05$ (thus, $KL(\dots)$ is evaluated only with first $j$ layers). Also note that weights of Lambda layers are shared across layers of the model.

### 3.1 Exit Criterion

During inference, Banino et al. (2021) proposed to sample the exit layer index from $p(i|x)$ (i.e., by sampling iteratively from a Bernoulli distribution with parameter $\lambda_i$). While a sampling-based exit criterion correlates with the variational view of PonderNet's training objective, such estimation introduces the randomness in the inference process of PonderNet.

| Method | | | | SST-2 | RTE | CoLA |
|---|---|---|---|---|---|---|
| ALBERT | | | | 92.7 ± 0.3 | 77.0 ± 1.9 | 57.0 ± 2.1 |
| ALBERT + PonderNet | | | | 91.1 ± 0.6 | 73.5 ± 1.9 | 50.8 ± 2.2 |
| ALBERT + PonderNet (Closed-Form Expectation) | | | | 92.3 ± 0.4 | 76.8 ± 3.0 | 55.9 ± 2.2 |
| Q-exit | Lambda LR | 3-Layer Lambda | $h$ concat. | | | |
| + | - | - | - | 92.2 ± 0.3 | 77.3 ± 1.4 | 55.7 ± 0.9 |
| + | + | - | - | 92.7 ± 0.4 | 77.3 ± 1.4 | 56.5 ± 1.2 |
| + | + | + | - | 92.6 ± 0.3 | 77.0 ± 1.4 | 56.3 ± 2.4 |
| + | + | - | + | **93.0 ± 0.3** | 76.5 ± 1.6 | 56.9 ± 1.9 |
| + | + | + | + | 92.9 ± 0.2 | **77.8 ± 1.2** | **57.4 ± 1.7** |

Table 1: An ablation study of the proposed PALBERT architecture. "Lambda LR" corresponds to fine-tuning the Lambda layer with its learning rate, "3-layer Lambda" refers to making the Lambda layer have three MLP layers instead of one, and "$h$ concat." stands for concatenation of two hidden states as input to the Lambda layer.

To overcome the issue of randomness, we propose **Q-exit**[2]: a novel deterministic criterion of performing an early exit, which we used for PALBERT. Instead of sampling from the distribution $p(i|x)$ during inference, we evaluate its CDF by accumulating $p(i|x)$ from each layer. Once the CDF is greater than the threshold hyperparameter $q$, we perform an early exit[3]. See Figure 1 for a schematic comparison of the sampling criterion with Q-exit. Threshold $q$ can be seen as a trade-off between underthinking and overthinking. Therefore, $q$ should be selected during the validation of the trained model to choose the best-performing value.

Based on our experiments, we found that the proposed criterion produced significantly better accuracy on various tasks compared to the original sampling criterion (see Sections 4.1, 4.2), while also being more practical than the original sampling criterion.

### 3.2 Lambda Layer Architecture

While the original PonderNet used a single layer MLP to obtain logit of exiting probability, we hypothesize that making the Lambda layer understand the dynamics of changing ALBERT hidden states is crucial for achieving good performance. To do so, instead of passing a single hidden state $h_i$ from the $i$-th layer in $\Lambda$, we concatenate it with $h_{i-1}$. I.e., for PALBERT, we evaluate the probability of exiting from $i$-th layer as $\lambda_i = \Lambda([h_i, h_{i-1}])$.

We used a 3 layer MLP with $\tanh$ activation for the Lambda layer to operate with more complex input. Based on the ablation study, we observed that increasing the capacity improves the accuracy of the trained model (See section 4.1). We also found it beneficial to fine-tune the Lambda layer with a different learning rate than all other parameters.

## 4 Experiments

### 4.1 Ablation Study

We performed an ablation study of the proposed changes in PonderNet architecture with PALBERT.

We experimented with adding the proposed Q-exit criterion, Lambda layer architecture, and fine-tuning strategies. These methods were benchmarked on SST-2, RTE, and CoLA tasks from the GLUE Benchmarking dataset (Wang et al., 2018).

We compared proposed changes with plain ALBERT and PonderNet adapted for ALBERT. Also, in these experiments we compared with PonderNet evaluated in expectation of predictions (e.g.,

---

[2]**Q**-exit stands for Quantile

[3]The proposed exit criterion is the simplest deterministic criterion which we came to, since there is no ability to estimate mean or argmax statistics during the inference without running all layers of the model, which is impractical if we want to perform an early exit.

| Method | SST-2 | RTE | QNLI | CoLA | MRPC | MNLI | QQP | STS-B | Macro |
|---|---|---|---|---|---|---|---|---|---|
| | | | | Dev set | | | | | |
| ALBERT | 92.7 | 76.5 | 91.5 | 56.6 | 90.5 | 84.8 | 88.9 | 90.6 | 84.0 |
| ALBERT (9 L.) | 91.1 | 74.2 | 91.2 | 56.9 | 89.7 | 84.3 | 89.7 | 90.1 | 83.4 |
| ALBERT PABEE | 92.7 | 76.9 | **91.5** | 55.6 | 88.3 | 84.5 | **88.9** | **89.9** | 83.5 |
| ALBERT PonderNet | 91.3 | 74.0 | 88.3 | 51.3 | 87.1 | 81.7 | 87.7 | 88.2 | 81.2 |
| PALBERT (ours) | **93.1** | **78.3** | 91.0 | **58.1** | **89.3** | **84.7** | 88.9 | **89.9** | **84.2** |
| | | | | Test set | | | | | |
| ALBERT | 93.4 | 70.0 | 92.1 | 50.5 | 85.6 | 79.0 | 84.7 | 87.4 | 80.3 |
| ALBERT PABEE | 92.7 | 71.1 | 91.3 | 46.0 | 84.3 | 79.2 | 83.7 | **86.5** | 79.3 |
| ALBERT PonderNet | 90.2 | 68.6 | 88.5 | 43.7 | 84.3 | 81.1 | 77.4 | 83.6 | 77.2 |
| PALBERT (ours) | **93.0** | **73.5** | **91.7** | **48.6** | **87.1** | 79.8 | **84.4** | **86.5** | **80.6** |

Table 2: A comparison of PALBERT with recent approaches on the GLUE benchmark. In the Macro column, we present the average results across tasks. We bolded the best results. Note that we did not bold the ALBERT rows since there is no early exit applied and we reported it for reference.

| Method | SST-2 | RTE | QNLI | CoLA | MRPC | MNLI | QQP | STS-B | Macro |
|---|---|---|---|---|---|---|---|---|---|
| | | | | Dev set | | | | | |
| RoBERTa | 94.5 | 79.4 | 92.4 | 63.1 | 91.3 | 86.7 | 89.8 | 90.7 | 86.0 |
| RoBERTa PABEE | **93.8** | 76.9 | **91.9** | 60.7 | 89.6 | 86.4 | **91.1** | **89.9** | 85.0 |
| PRoBERTa (ours) | 93.7 | **79.1** | 91.6 | **62.2** | **90.2** | **86.7** | **91.1** | **89.9** | **85.6** |
| | | | | Test set | | | | | |
| RoBERTa | 94.5 | 74.1 | 93.1 | 58.9 | 89.1 | 86.5 | 80.9 | 88.4 | 83.2 |
| RoBERTa PABEE | **95.4** | 71.7 | **92.0** | 51.2 | 87.6 | **86.2** | **80.4** | **86.2** | 81.3 |
| PRoBERTa (ours) | 94.5 | **72.1** | 91.8 | **58.6** | **88.3** | 86.1 | 79.6 | 85.7 | **82.1** |

Table 3: A comparison of PRoBERTa with recent approaches on the GLUE benchmark. In the Macro column, we present the average results across tasks. We bolded the best results. Note that we did not bold the RoBERTa rows since there is no early exit applied and we reported it for reference.

evaluated $p(y|x) = \mathbb{E}_{i \sim p(i|x)}\Big[p(y|x, i)\Big]$ in closed-form). While such a model is impractical for early exiting setup (it does not perform any early exit but evaluates all layers to estimate the expectation), it could show a gap between closed-form expectation of model predictions and its single sample Monte Carlo estimation.

For evaluation, we performed a grid hyperparameter search on an appropriate metric score on the dev split for each dataset. Following the PABEE training setup, we trained all models with a fixed learning rate until validation metrics stopped increasing for 5 epochs. We used Adam optimizer (Kingma and Ba, 2015) for all experiments, a fixed $q = 0.5$ on models with the Q-exit criterion, as well as a fixed classifier dropout value equal to $0.1$ (Srivastava et al., 2014), and $\lambda = 0.1$.

We trained each model 5 times with the best hyperparameters and reported the mean and std values. A full list of the methods' hyperparameter ranges can be found in Table 4.

See Table 1 for the full list of the results of our ablation study. Based on these experiments, PonderNet architecture is seen as performing worse than vanilla ALBERT fine-tuning. Also, PonderNet with closed-form evaluation of predictions expectation performed better than the sampling criterion, indicating that randomness in exit layers leads to poor performance.

At the same time, the deterministic Q-exit criterion improves PonderNet accuracy when compared to a random sampling of the exit layer and is comparable to the evaluation of PonderNet in expectation. A more complex Lambda layer that can handle the dynamics of hidden state changes can further improve model accuracy when compared to the original PonderNet.

## 4.2 GLUE Experiments

We compared PALBERT with PonderNet architecture adapted for ALBERT fine-tuning, and PABEE with a fixed patience value $t = 6$ on all GLUE tasks. We also trained Ponder RoBERTa (PRoBERTa) and compared it with PABEE and adapted for RoBERTa. Note that we re-implemented all baselines used in our experiment for consistency in comparison. See Appendix Section A for the details of reimplementation.

Following the experimental setup from Section 4.1, we trained 5 models with the best hyperparameters across the hyperparameter search and reported the median task score on the dev set. We evaluated the test scores on the best models, selected based on their dev scores. We report the two metrics' mean for the MRPC, QQP, and STS-B tasks. For the MNLI task, we report the mean accuracy across matched and mismatched datasets.

See Tables 2,3 for the full list of results. We observed that PALBERT outperformed PABEE on a wide range of tasks. Vanilla PonderNet with the sampling exit criterion performed the worst. Vanilla ALBERT outperformed PABEE on most tasks and is comparable to PALBERT, while the latter has the highest score averaged across all tasks (see Macro column in Tables 2, 3).

For experiments with RoBERTa, we observed that PRoBERTa outperformed PABEE on Dev split, while it either performed better or marginally worse for the Test set. Thus, PRoBERTa showed a higher macro score compared to PABEE.

Note that PABEE is performing poorly for tasks with a small dataset (e.g., CoLA, RTE). We hypothesize that this is caused by several independent classifiers at each layer $C_i$ failing to train well enough, whereas PALBERT was capable of utilizing knowledge sharing between layers.

| Parameter | Values range |
|---|---|
| Learning rate | [1e-5, 2e-5, 3e-5, 5e-5] |
| Batch size | [16, 32, 128] |
| Lambda learning rate | [1e-5, 2e-5, 3e-5] |
| $\beta$ | [0.5] |

Table 4: Hyperparameter search ranges used in all of our experiments.

## 4.3 Understanding the Threshold of Q-exit

As noted previously in Section 3.1, we treat the threshold value $q$ of the Q-exit criterion as a trade-off between underthinking and overthinking, where increasing $q$ forces a model to evaluate more layers, and vice versa.

Therefore, it is necessary to find the best-performing threshold for each task where a model has the highest accuracy. To do so, we evaluated trained PALBERT models from the ablation study (see Section 4.1) on dev splits of tasks with different values of $q$. We then averaged obtained metrics and reported the mean and std values for various thresholds (See Figure 2 for the results).

We observed that exiting models with $q = 0.5$ shows the best overall performance for different tasks. Making $q$ greater than $0.5$ leads to a reduction in accuracy and can often force models to evaluate all 12 layers of ALBERT-Base.

It is also notable that PALBERT, having a large threshold value $q$ that performs constant exit on the last layer, has better accuracy than vanilla ALBERT fine-tuned for the SST-2 task. This indicates that adding auxiliary tasks for each layer of ALBERT improves performance compared to plain fine-tuning with a single classifier on the last layer of the model.

## 4.4 Speed Analysis

While making $q < 0.5$ improves inference speed, it can also lead to underthinking and lower accuracy (see Figure 3). We compared PALBERT using different threshold values $q$ to PABEE with different patience values $t$, which stands for the number of layers necessary to output the same result in a row to perform an early exit. We evaluated task scores for the specified hyperparameters as well as the increase in speed when compared to vanilla ALBERT inference of a full model with 12 layers.

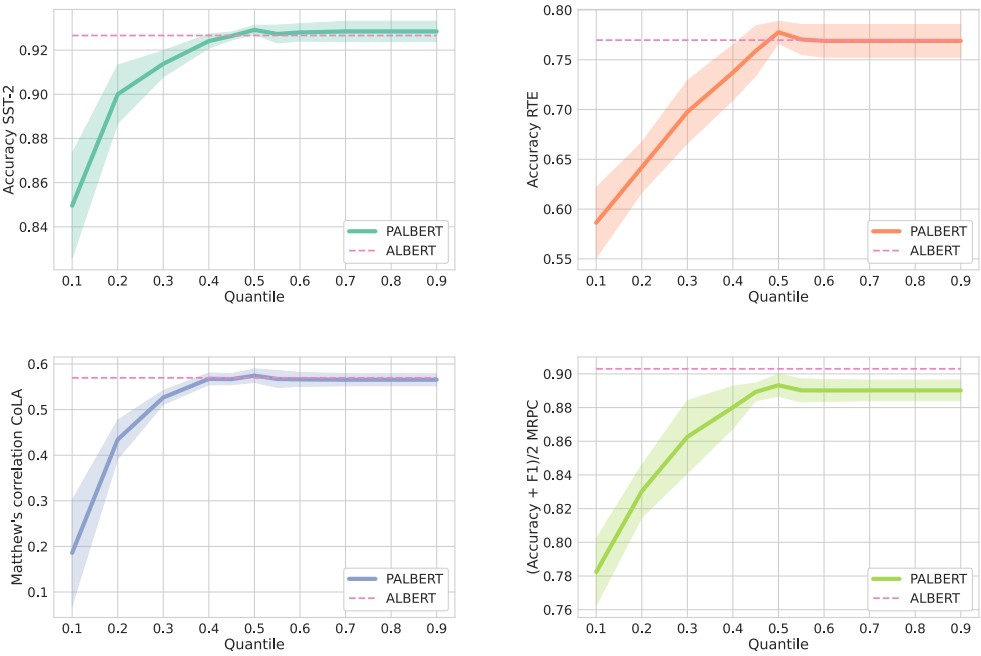

Figure 2: PALBERT score dependency on the Q-exit threshold. We report the mean and std values of task metrics across 5 trained models. See section 4.3 for more details

Overall, we observed that PALBERT mostly produced higher scores on different tasks while also being slightly faster than PABEE. For the CoLA and MRPC datasets, PALBERT performed better. The proposed method outperformed PABEE while achieving the same increase in speed.

We observed questionable results on the SST-2 dataset: the best score for the PABEE model is slightly higher than that of PALBERT. However, it was obtained with a negligible increase in speed compared to ALBERT, as the best-performing patience for this setup is 11 layers while the whole model has only 12 layers.

Furthermore, unlike PALBERT, PABEE performed significantly worse than plain ALBERT fine-tuned on the CoLA and RTE tasks.

## 4.5 On the Prior of PALBERT

While we used a fixed value of $\lambda = 0.1$ in our main experiment following Banino et al. (2021) (see Section 4.2), we also investigate the effect of changing the parameter of the prior distribution on exit layer indices.

We followed the experimental setup from the previous sections and trained models with $\lambda \in [0.1, 0.15, 0.25, 0.5]$. For each $\lambda$, we had its own best hyperparameter set, which was then used to train 5 models.

See Figure 4 for the results. We observed that varying $\lambda$ affects the performance of the trained model since it has a huge impact on the distribution of exit layer indices. Although the best metric for the RTE task was reached with $\lambda = 0.08$, such a model mostly performs an exit from the last layer. Meanwhile, $\lambda = 0.08$ performs marginally worse than $\lambda = 0.1$ for MRPC.

Note that while Figure 4 was obtained using the Q-exit criterion, we could also estimate $\mathbb{E}_{x \sim D}\left[p(i|x)\right]$ (see Figure 5), which is the distribution of exit layer indices with vanilla sampling criterion proposed by Banino et al. (2021). Even if $\lambda$ is low enough so that the mode of distribution is the 12-th layer, in almost half of the cases, the sampling criterion forces the model to exit from one of the first 10 layers. We hypothesize that such randomness in the index of the exit layer leads to the poor performance observed during previous experiments (see Section 4.2).

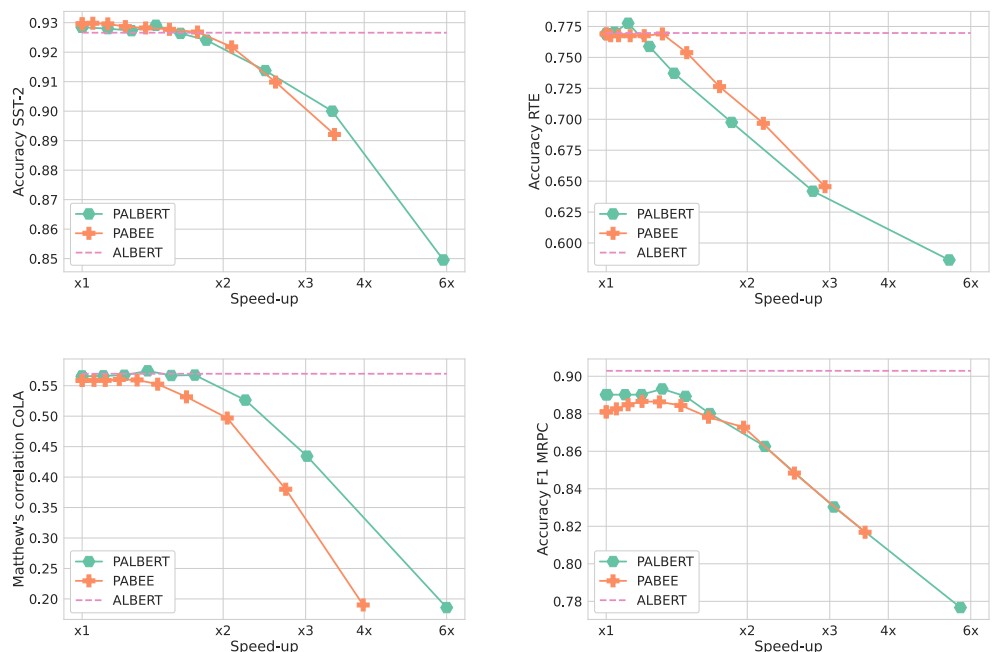

Figure 3: A comparison between PALBERT and PABEE models on SST-2, RTE, CoLA, and MRPC tasks. We varied the threshold value of Q-exit for PALBERT and the patience hyperparameter for PABEE to obtain the plots of task scores of inference increasing in speed. 1x stands for plain ALBERT inference without performing an early exit. The horizontal line corresponds to plain ALBERT fine-tuning. See Section 4.4 for the analysis of these plots.

From such perspective, dependency on the explicit prior distribution could be seen as the main limitation of the proposed method, since treating the exit layer index as a latent variable justified its prior distribution. If proper prior distribution was found, then the well-performing model could be trained, and vice versa.

## 5   Future Work and Limitations

In this paper, we proposed improving the PonderNet architecture to perform an early exit using fine-tuned ALBERT and RoBERTa models with the novel Q-exit criterion and a revisited Lambda layer architecture. This approach is orthogonal to the consensus-based approaches (Zhou et al., 2020; Zhu, 2021) highly represented in the field.

While PALBERT and PRoBERTa outperformed some recent State-of-The-Art methods used for an early exit, there is a clear direction for further improvement of this method, as it was not capable of outperforming plain ALBERT and RoBERTa on some GLUE tasks.

We believe that PALBERT could benefit from the development of a new parameterization of the prior distribution on exiting from each layer since it directly affects the resulting posterior distribution used to perform an early exit (see Section 4.5). However, the development of an appropriate prior distribution for early exiting models is orthogonal to the development of a proper exit criterion.

Furthermore, there is no theoretical justification for the Q-exit threshold value. Although we observed that $q = 0.5$ performed best, it is without a clear explanation as to why that is so. We hypothesize that bringing more insights into developing deterministic exit criteria could further improve the proposed method.

In addition, adding more auxiliary tasks could also make it possible to improve PALBERT further. This way, PALBERT training can be made more PABEE-like by making independent classifiers for each layer of the model or adding self-distillation across layers.

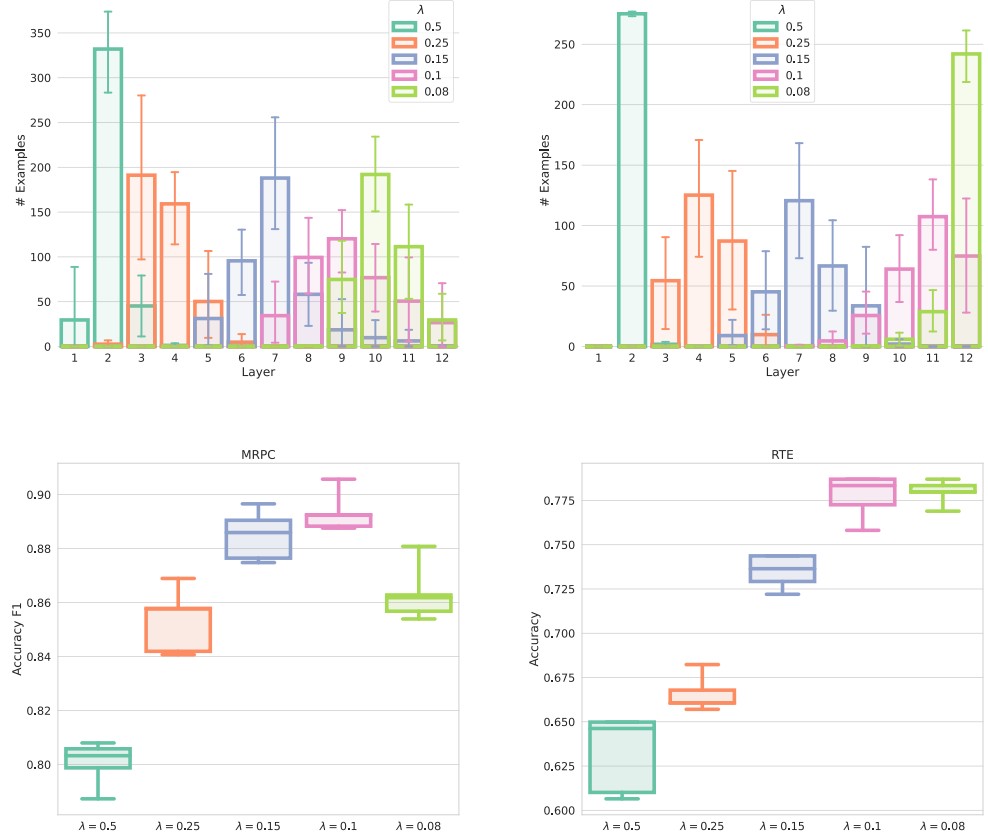

Figure 4: Histogram of exit layer indices with Q-exit criterion and different prior distribution parameters $\lambda$ (top) alongside with task metrics for trained models (bottom) for MRPC and RTE tasks. See Section 4.5 for more details.

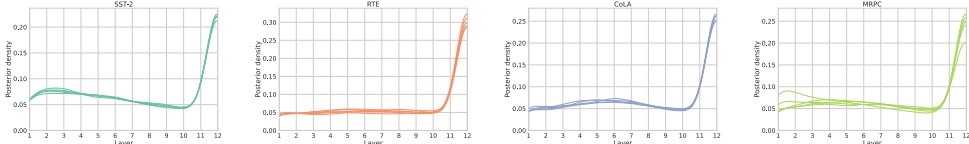

Figure 5: An estimation of $\mathbb{E}_{x \sim D}\left[p(i|x)\right]$, where $p(i|x)$ is a trained posterior probability of exiting from layer $i$ of PALBERT models across different tasks, $\lambda = 0.1$, and $D$ is the distribution of the training dataset. We took 5 models trained on these tasks and sampled exit layer indices for the training dataset's inputs. We smoothed the obtained probabilities for visibility. Note that these probabilities could be seen as the distribution of exit layers across different tasks for plain PonderNet with sampling criterion, which indicates on high variance in indices of output layers.

While the variational view of the early exiting mechanism allows us to train models which could be exited early, we observed that the overall performance of the trained model highly depends on the prior distribution of exit layer indices (see Section 4.5). Because of this, the proposed method requires more computation for hyperparameter search compared to PABEE (Zhou et al., 2020). We believe that, in order to further develop training-level early exiting, it may be beneficial to move away from variational inference to some other form of trainable mechanism without relying on exit layers' explicit prior knowledge.

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
