# A    Re-implementation of Baselines

While Zhou et al. (2020) provided official implementation of the proposed method, we have several concerns regarding its reproducibility and experimental setup. The original implementation implies running evaluation iteration-wise (i.e., the best performing model is selected once in $n$ iterations).

In this case, several iterations $n$ between validation epochs could be seen as an additional hyperparameter, which has a large impact on the resulting performance. Although, Zhou et al. (2020) did not include the best performing value so that the results could be easily reproduced.

Because of this, we decided to re-implement PABEE strictly following the original paper and training details, while evaluating the model after every epoch, instead of evaluating after $n$ training steps, which we believe is a fairer way to compare methods.