# OpenReview forum: "PALBERT: Teaching ALBERT to Ponder"
_NeurIPS.cc/2022/Conference — NeurIPS 2022 Accept_

### Official Review · Reviewer_yzdP · 2022-06-20

**Rating:** 4
**Confidence:** 5
**Soundness:** 2 fair
**Presentation:** 2 fair
**Contribution:** 2 fair

**Summary:**

This paper focuses on early exit mechanism for encoder-only Transformer models. Specifically, it uses the ALBERT architecture that is similar to BERT but shares parameters across all layers (12 layers for base size).

The authors adopt the PonderNet framework of Banino et al. that learns early-exit "lambda" classifiers for each layer, and suggest two modifications to it. First, instead of sampling from the geometric-like distribution that PonderNet emulates to decide if to exit, they compute the empirical CDF of the layers so far, and exit if the value is greater than some defined threshold q. Second, they change the module that computes lambda to be slightly larger and to use the last two hidden states instead of just the current one, and modify the learning rate.

Experiments are completed on the GLUE tasks, where PonderNet and PABEE (exiting if several consecutive layers predict the same) are used as baselines (authors reimplemented these baeslines to use the same ALBERT backbone).  Also, a grid-search over threshold values for q and ablation studies on hyper-parameters are reported.

**Questions:**

See points above, also:
* Why PonderNet in table 2 doesn't have test results?
* How were the hyper-parameters and threshold decided for each model in table 2?

**Limitations:**

I don't see potential negative societal impact of this work.

Limitations are described in the last section.

**Strengths And Weaknesses:**

Strengths:

1. The paper deals with an important and impactful task of improving the efficiency of pretrained Transformer-based NLP classifiers
2. Empirical results suggests that the proposed modifications to PonderNet could be effective, though I have concerns on the evaluation and comparisons (see below)
3. The writing is overall clear and easy to follow, though there are several unsupported claims (see below)
4. The ablation studies could help future studies

Weaknesses:


1. The authors claim that other methods are heuristics, but the suggested method is also a heuristic with no theoretical grounding (as the authors admit in section 5). In fact, sampling from the geometric-like distribution seems to me more grounded as it aims to simulate the probability of a "good exit" exactly at the current layer, compared to the CDF that computes the probability that one of the layers so far has been a "good exit" layer, but the prediction is taken from the current layer, so the chance of an earlier exit being good is not relevant anymore. Perhaps the authors can clarify this?
2. The authors main motivation for using CDF instead of sampling from the geometric-like distribution is having a deterministic exiting rule (e.g. see bolded span lines 39, and lines 118, 126-127). However, I wasn't convinced (1) why determinism is better, and (2) why the CDF is needed for achieving determinism. Assuming I had a calibrated predictor [1], I don't see a particular issue with sampling as the training and evaluation are anyway in expectation. Also, why not make a deterministic threshold-based decision directly on the lambda values like in [2]. Though, to obtain theoretical-backed guarantees on the marginal performance of the model, there should be some MHT correction, or other calibration process for the threshold value like in [3].
3. The main evaluation in table 2 presents a different tradeoff point between the models. The accuracy is bolded as being better, but the speedup for PALBERT is lower (also, how is the speedup computed is missing). It is impossible to compare the models this way as it is unclear how does the performance compare when tuned for the same speedup or vice versa. For example, figure 3 shows at least for SST-2 several points where PABEE is better than PALBERT, but in table 2 PALBERT is reported with higher accuracy.
4. No details on how the reported speedup value is computed. Does it take into account the computation of the lambda function (that is now larger)?

[1] On Calibration of Modern Neural Networks

[2] Depth-Adaptive Transformer

[3] Consistent Accelerated Inference via Confident Adaptive Transformers

**update post author response**: I appreciate the author's detailed response. I've decided to slightly increase my score. I'm still not entirely convinced about the motivation behind the "deterministic" exiting rule. But more importantly, I think that a table with only accuracies is insufficient for evaluating such early exiting models since there is some tradeoff with the actual speedup of each model. According to figure 3, the gains over PABEE are inconsistent across tasks.

---

> ### Author Response · Authors · 2022-08-01
> **Reviewer yzdP answer**
>
> The reviewer proposed an alternative exit criterion based on values of lambdas during the inference. Also, there are concerns about the proposed exit criterion as it is less grounded than the original sampling criterion.
>
> Question: In fact, sampling from the geometric-like distribution seems to me more grounded...
>
> We admit that sampling from the posterior could be seen as more grounded, and from such a perspective, we should not claim our method not to be heuristic. We will fix our claims in the next revision of the paper. Still, while sampling is more grounded, it is impractical and leads to lower results (which we observed in Section 4.1).
>
> Question: I wasn't convinced (1) why determinism is better
>
> We provided quantitative evidence that determinism performed better than the sampling criterion with the Ablation study. While the sampling criterion is more grounded when the expectation is evaluated in closed-form, in practice, it could be only estimated with a single sample, which leads to high variance in exit layer indices (see Figure 5)
>
> Question: why the CDF is needed for achieving determinism
>
> We desired to provide the most straightforward deterministic criterion. While a criterion based on values of lambdas seems interesting for further research, it requires ensuring that model predictions are calibrated, which seems to be more complicated than the proposed criterion.
> Also, there is no apparent reason why different ways of calibration would be superior to the exit criterion. The calibration of neural networks is an important research direction. However, an early exit is an orthogonal way to think about this problem.
>
> Question: It is impossible to compare the models this way ...
>
> We performed a hyperparameter search for each method (L162). While ALBERT and PonderNet do not have any threshold values, for PABEE we used a fixed patience threshold (L159) to evaluate Table 6 since the original work reported this value as best performing. For PALBERT we also used a fixed threshold q=0.5 since it was found as best performing in Section 4.3. Overall, these thresholds were selected as best performing across all tasks to evaluate the Test set with them.
> We used Figure 3 to provide more detailed insights on the performance of PALBERT compared to PABEE with different thresholds, which is an orthogonal way to compare these models. We will add other GLUE tasks to this Figure in the final version of the paper.
>
> Question: No details on how the reported speedup value is computed. Does it take into account the computation of the lambda function (that is now larger)?
>
> Thank you for pointing this out. We will add these details to the paper. Answering the question, we took into account all computations. We measured the wall-clock time of model inference to evaluate the speed-up.
>
> Question: Why PonderNet in table 2 doesn't have test results?
>
> Thanks for pointing out this critical issue that we underemphasised. Original PonderNet performed significantly worse based on the Ablation study and Dev set results. Because of this fact, we decided to concentrate on comparison with PABEE and provide more experiments with it. Although, we agree with this concern and will add Test results of PonderNet in the final version of the paper for completeness.

---

### Official Review · Reviewer_Ndy3 · 2022-07-07

**Rating:** 6
**Confidence:** 3
**Soundness:** 3 good
**Presentation:** 4 excellent
**Contribution:** 2 fair

**Summary:**

This paper aims to improve the inference time efficiency of ALBERT, a pretrained large transformer, through the early exit. In particular, a dedicated binary classifier is trained to decide, at each layer, whether or not to use the current layer’s output as the final representation and skip further forward propagation. If the model exits at an early layer, significant efficiency gains can be achieved. This work heavily builds on a previous work along this line—the PonderNet—and improves its test-time exit criterion: instead of drawing from a Bernoulli, this work proposes to accumulate the early-exit classifier’s output probability throughout the layers and determine the early exit by comparing it to a prespecified threshold. The motivation is to reduce the variance. This method is combined with other techniques aiming to improve the capacity of the early-exit classifier. The model is evaluated on the GLUE benchmark, with extensive ablation study, tradeoff curves, and other analyses. The proposed model outperforms previous early-exit methods designed for ALBERT, and improves the efficiency with little or no accuracy drop.

I find the idea very intuitive and clearly presented. The detailed analysis and honest results are a big plus. However, I do have concerns about the technical contributions. It seems that most of the proposals heavily build on previous works and are rather engineering. Maybe the authors could help me and clarify their technical contributions in the response. Further, the method is presented in a way that can only be used in ALBERT. Exploring its applications in other pretrained models could improve its impact.


**Questions:**

- It may happen that different instances in a mini-batch exit at different layers. How is this resolved? Would it affect the efficiency?
- The paper argues that a major drawback of PonderNet is that it has a higher variance. But the std is barely larger than ALBERT in Table 1. Could the authors comment on this? Also, it would be nice to include std too in Table 2.
- It would be interesting to see how the exit layers might correlate with the inputs’ lengths, labels, or some other metrics measuring their difficulty.
- Have the authors compared to a handcrafted exit strategy? One can dedicate that the model must exit after X layers to achieve the 1.29x speedup, and compare its accuracy with learned exit strategies.

**Limitations:**

The authors have discussed the limitations in detail. I don't have any further concerns about the potential negative societal impact.

**Strengths And Weaknesses:**


Strengths:
- Very clear writing
- Very detailed and informative analysis of the results.
- The method is intuitive.
- Honest and detailed discussion of limitations

Weaknesses:
- Early exit methods have been explored in other large pretrained models (e.g., [1]). Applying Ponder ALBERT in other pretrained models could improve its impact.
- The 1.29/1.26x speedup without sacrificing accuracy is marginal.

[1] https://arxiv.org/abs/2004.07453

---

> ### Author Response · Authors · 2022-08-01
> **Reviewer Ndy3 answer**
>
> The reviewer proposed more experiments to include in the paper. Also, there is concern about the performance of the proposed mechanism with batched inference.
>
> Question: It may happen that different instances in a mini-batch exit at different layers. How is this resolved? Would it affect the efficiency?
>
> Different methods for handling this issue could be applied. E.g., one could wait for the last instance in the mini-batch to be evaluated, while other instances could be dropped and thus reducing the utilization of GPU during the evaluation.
> Considering the real-world setting, the way the model could handle multiple instances in a batch highly depends on the RPS of the application, as well as the number of available GPUs. In the general case, this could slightly affect the efficiency of any model performing early exit. But there could be specific cases when the optimal batch size equals 1.
> However, we would like to note that this question refers not just to the PALBERT but to the research direction itself. While this question is important, we believe that answering it is beyond this paper.
>
> Question: The paper argues that a major drawback of PonderNet is that it has a higher variance...
>
> There are two different variances. The first is the variance of the model's score on a specific dataset. The second (referred to in L11, L45) is the variance of the exit layer index). Quantitative evidence of variance in exit layer indices for plain PonderNet is depicted in Figure 5. We also confirm that randomness in the exit criterion reduces the accuracy of trained models with the Ablation study (L145).
>
> Also, we had these std values in Table 2 in the early version but omitted them later for visibility. Thank you for pointing out this issue, we will add them in the final version of the paper.
>
> Question: It would be interesting to see how the exit layers might correlate with the inputs’ lengths, labels, or some other metrics measuring their difficulty.
>
> Thank you for proposing this idea. We did perform such an experiment using entropy predictions as a measurement for the difficulty of samples. We observed that both PABEE and PALBERT exit layer indices correlated with the complexity of samples. There were no interesting insights into the performance of these methods and thus we decided to omit them in the paper to highlight other results.
>
> Question: Have the authors compared a handcrafted exit strategy? One can dedicate that the model must exit after X layers to achieve the 1.29x speedup, and compare its accuracy with learned exit strategies.
>
> We will definitely add such an experiment in the final version of the paper.

---

> > ### Comment · Reviewer_Ndy3 · 2022-08-03
> > **Thanks for the response**
> >
> > I'd like to follow up on the variance of PonderNet. Thanks for clarifying the std in Table 1. In addition to Figure 5, could the authors quantify the impact on accuracy of PonderNet's high variance in exit indices? A good measure would be the std across multiple test runs of the same PonderNet model. I think this is important to back up a central claim of the paper. It could also help address other reviewers' concerns on choosing a deterministic strategy over a stochastic one.
> >
> > Besides, I agree with reviewer RdPQ that testing out PALBERT on other tasks/pretrained models would be a big plus.

---

> > > ### Author Response · Authors · 2022-08-03
> > > **Reviewer Ndy3 answer**
> > >
> > > Thank you for proposing the idea with std across multiple test runs. We are adding PonderNet with a closed form expectation of predictions across layers to the paper right now, so it will explain the issue with the sampling criterion. With the proposed idea our claim will be even stronger.

---

### Official Review · Reviewer_RdPQ · 2022-07-11

**Rating:** 3
**Confidence:** 4
**Soundness:** 3 good
**Presentation:** 2 fair
**Contribution:** 2 fair

**Summary:**

Previous work found that the computation of a deep neural network can exit at earlier layers to prevent "overthinking". This improves upon previous method (PonderNet) of performing an early exit and proposes a new deterministic Q-exit criterion. This new method determines the exit layer index in a deterministic way by using the CDF of the layer exit distribution. Experiments of language understanding tasks (GLUE) and demonstrate improved performance over PonderNet.

**Questions:**

The questions are asked in the weaknesses section.

**Ethics Review Area:**

["I don’t know"]

**Limitations:**

N/A.

**Strengths And Weaknesses:**

**Strengths:**
1. The paper is relatively well written and proposes a simple and effective method to alleviate the high-bias issue of sampling based layer exit method.
2. Comprehensive analysis have been conducted to show the effects of the hyperparameter of threshold, the speedup and the distribution of examples that exit at different layers.

**Weaknesses:**
1. The original PonderNet paper which this paper is based on is evaluated on more complex tasks, e.g. QA and other multi-step reasoning tasks, which I find a more appropriate setup for evaluating this layer-exit approach, which is a better test bed for these approaches. With that being said, experiments on the GLUE task alone is not sufficient to demonstrate the effectiveness of the proposed method. Moreover, as discussed in the limitation section, the proposed method still lags behind the Albert model on some tasks, I think this paper still needs further improvements for a better contribution.
2.  This paper uses Albert model as the base model for executing the proposed method which is a limitation of the proposed method. I was wondering if there is a specific reason for the authors not to use other pre-trained models, especially deeper networks, to implement the idea. If this is because Albert uses shared weights for layers, then is the proposed method a model-specific trick but not a generic method?
3. The threshold p that decides when to exit seems to be a very important hyperparameter for the success of the proposed method and it is selected with the validation set. This directly contrasts with selecting the best exiting layer from 1 to 12 layers as a baseline. This simple baseline is not included in the paper. Could you also compare with that?
4. A suggestion on the presentation of the paper: it's better to describe Albert and PonderNet more clearly in a background section to make the paper more clear to the general audience.

---

> ### Author Response · Authors · 2022-08-01
> **Reviewer RdPQ answer**
>
> Thank you for the review. We consider adding experiments with QA and other architectures in the final version of the paper.

---

### Official Review · Reviewer_id4o · 2022-07-12

**Rating:** 6
**Confidence:** 3
**Soundness:** 3 good
**Presentation:** 2 fair
**Contribution:** 2 fair

**Summary:**

This work proposes an improvement over PonderNet, an early-exit method that can help overcome long inference times and "overthinking" in a neural network. The specific improvement is the change of exit layer selection during inference-time, using a deterministic criterion instead of the probabilistic method proposed originally in PonderNet. The paper compares the improvements to both PonderNet (partially) and PABEE, a heuristic based early-exit method. The results are presented using ALBERT and a variety of GLUE tasks. Overall, the work shows that there are small but consistent gains over earlier methods, with a 1.2-1.3x speedup in inference compared to vanilla ALBERT.



**Questions:**

Table 1 shows the overall variance of accuracies across several tasks. While the variance of PonderNet is slightly higher than PALBERT, it does not justify the claim that there is "major variance" (L11, L45) in PonderNet's outputs. Do you have some quantitative evidence for this claim?

Line 159 mentions that the baselines were re-implemented, but PonderNet results are missing from the test set section in Table 2. What is the reason for these missing results?
Figure 3's caption says that the results are for two tasks, while the results for 4 tasks are presented, please fix this.

The paper repeatedly mentions that PALBERT "significantly outperforms" or "outperforms by a large margin", while the actual overall improvements are not qualified by this. Consider simplifying the language and/or presenting more evidence to support this claim.
L160 "Ablation" -> "Appendix"

**Limitations:**

Authors have addressed limitations adequately.



**Strengths And Weaknesses:**

Strengths

Consistent gains over previous methods
Logical improvements over previous work (using two layers to study change across them)
Evaluation of the method itself is done well (ablation studies for the sub-component across a few tasks, speed comparison with one other method)

Weaknesses

Paper is not very clearly written, and required several passes to fully grasp.
It seems like the PonderNet results were added after the fact (e.g. they are missing from the speed vs performance graph, missing test set results)
The work does not shed much light on why PALBERT works better, and seems like a more empirically motivated and studied method. While there is nothing wrong in this in an itself, the improvements themselves are fairly incremental over PonderNet, and may not surpass the base quality of this particular venue.

---

> ### Author Response · Authors · 2022-08-01
> **Reviewer id4o answer**
>
> There is a concern that the proposed method is incremental compared to the previous works, which may not fit the NeurIPS venue. The reviewer points out the lack of experimental results with the original PonderNet. Also, there are questions on paper writing.
>
> Question: While the variance of PonderNet is slightly higher than PALBERT, it does not justify the claim that there is "major variance" (L11, L45) in PonderNet's outputs...
>
> There are two different variances. The first is the variance of the model's score on a specific dataset. The second one (referred to in L11, L45) is the variance of the exit layer index. Quantitative evidence of variance in exit layer indices for plain PonderNet is depicted in Figure 5. We also confirm that randomness in the exit criterion reduces the accuracy of trained models with the Ablation study (L145).
> We will clarify the writing in the final revision of the paper.
>
> Question: PonderNet results are missing from the test set section in Table 2
>
> Thanks for pointing out this important issue that we underemphasised. Original PonderNet performed significantly worse based on the Ablation study and Dev set results. Because of this fact, we decided to concentrate on comparison with PABEE and provide more experiments with it. Although, we agree with this concern and will add Test results of PonderNet in the final version of the paper for completeness.
>
> Concern: Consider simplifying the language...
>
> We will fix our claims in the final version of the paper.
>
> Concern: The work does not shed much light on why PALBERT works better, and seems like a more empirically motivated and studied method. While there is nothing wrong in this in an itself, the improvements themselves are fairly incremental over PonderNet, and may not surpass the base quality of this particular venue.
>
> While the original motivation for improvements of PonderNet was empirically motivated, we found quantitative evidence of the poor performance of sampling exit criterion compared to deterministic criterion, which we addressed in the Ablation study, as well as in other experiments (e.g., Section 4.5).
>
> Also, we would like to highlight that while current results may seem incremental, these results could be used as a foundation for further research since the proposed method is orthogonal to widely adopted recent methods. From a certain perspective, one could have thought that the variational view of training an early exit mechanism performs poorly compared to PABEE. However, this paper showed that non-conventional ways to solve this task could be applied with carefully chosen exit criterion.

---

### Author Response · Authors · 2022-08-09
**Paper Improvements**

We have addressed the weaknesses discussed in the review and performed experiments with the most important ones. We uploaded new results to the rebuttal revision of the paper and kindly ask reviewers to consider them.

1) Reviewers id4o and yzdP were concerned about the importance of the deterministic exit criterion proposed in the paper.

To address this issue, we trained a PonderNet model for which we performed inference in expectation across layers. Treating the sampling criterion as a single sample Monte Carlo estimation of this expectation, we could show that such an estimation leads to poor performance.

We trained PonderNet, which was evaluated with the expectation of predictions across layers (we trained a new model to select the best performing checkpoint with a new criterion) with datasets from the ablation study and got the following results:

- SST-2: 92.3±0.4,
- RTE: 76.8±3.0,
- CoLA: 55.9±2.2

PonderNet, in expectation, outperformed the sampling criterion by a large margin, indicating that randomness in the exit criterion is a possible reason for poor performance, which matches our points in the paper. Also, note that these results are comparable with the Q-exit criterion in the ablation study.

2) Reviewers id4o and yzdP pointed out missed PonderNet test results.

We evaluated them and added them to the paper.

3) Reviewer RdPQ proposed comparing the handcrafted exit strategy with a fixed exit layer.

We added an experiment with ALBERT being trained to exit from the 9-th layer.

4) Reviewers id4o, RdPQ, Ndy3, and yzdP proposed to evaluate the proposed mechanism with other than ALBERT models.

We trained RoBERTa-Base with the proposed pondering mechanism and compared it with plain RoBERTa-Base and RoBERTa-Base with the mechanism proposed with PABEE.

Pondering RoBERTa either outperformed or performed on pair with PABEE with Dev set. Also, it outperformed Pondering ALBERT, thus showing that the proposed method is not only applicable to ALBERT.

---

### Meta-Review · Area_Chair_hwYF · 2022-08-29

**Recommendation:** Accept
**Confidence:** Less certain

**Metareview:**

This is an interesting paper which improved during discussion and the authors in my opinion addressed many of the suggestions and concerns from all of the reviewers (two reviewers who engaged after significant discussion were supportive of acceptance).  I personally think that the paper will be an interesting addition to neurips but could be stronger if experiments were performed beyond just the GLUE/SuperGLUE setup and considered other complex inference tasks such as question answering (e.g. natural questions).

**Award:**

Yes

---

### Decision · Program_Chairs · 2022-09-14

Accept